# Common SNCA Genetic Variants and Parkinson’s Disease Risk: A Systematic Review and Meta-Analysis

**DOI:** 10.3390/ijms26136001

**Published:** 2025-06-23

**Authors:** Raziyeh Mohammadi, Mahdi Shirazi, Sayedeh Fatemeh Sadat-Madani, Matthew Zachary Yeo Cheng Long, Corrine Lee Singh, Jayne Y. Tan, Xiao Deng, Seyed Majid Hashemi Fard, Samuel Y. E. Ng, Adeline S. L. Ng, Louis C. S. Tan, Seyed Ehsan Saffari

**Affiliations:** 1Duke-NUS Medical School, National University of Singapore, Singapore 169857, Singapore; raziyeh.mohammadi@duke-nus.edu.sg (R.M.); hashemi@nus.edu.sg (S.M.H.F.); adeline.ng.s.l@singhealth.com.sg (A.S.L.N.); louis.tan.c.s@singhealth.com.sg (L.C.S.T.); 2Department of Physics, Isfahan University of Technology, Isfahan 84156-83111, Iran; mehdi.shirazi@alumni.iut.ac.ir; 3School of Medicine, Isfahan University of Medical Sciences, Isfahan 81746-73461, Iran; dr.sadatmadani@gmail.com; 4Ministry of Health Holdings (MOHH), Singapore 139691, Singapore; matthewzachary.yeo@mohh.com.sg; 5Yong Loo Lin School of Medicine, National University of Singapore, Singapore 117597, Singapore; e0771290@u.nus.edu; 6Department of Neurology, National Neuroscience Institute, Singapore 308433, Singapore; jayne.tan.yi@singhealth.com.sg (J.Y.T.); deng.xiao@sgh.com.sg (X.D.); 7Department of Research, National Neuroscience Institute, Singapore 308433, Singapore; samuel.ng.y.e@singhealth.com.sg

**Keywords:** alpha-synuclein, neurodegenerative disorders, personalized medicine, heterogeneity

## Abstract

The *SNCA* gene, encoding alpha-synuclein, is implicated in the pathogenesis of Parkinson’s disease (PD), with several single-nucleotide polymorphisms (SNPs) linked to increased risk. This study systematically evaluated the association between common *SNCA* polymorphisms and PD through a meta-analysis of cohort and case–control studies published before 20 November 2023. Eligible studies were identified via comprehensive searches of PubMed, Scopus, and Web of Science, and pooled odds ratios with 95% confidence intervals were calculated under allelic, dominant, and recessive models. Heterogeneity and publication bias were assessed, and subgroup and sensitivity analyses were performed. Twenty-seven studies were included. SNP *rs11931074* showed consistent associations with PD across all models, with low heterogeneity and no evidence of publication bias. rs356219 and rs356165 were also significantly associated with PD, although regional differences contributed to heterogeneity. In contrast, rs2583988 showed marginal significance in the allelic model, which was lost after sensitivity analyses. No associations were found under dominant or recessive models for this SNP. These findings confirm rs11931074 as a robust PD risk variant and support the roles of rs356219 and rs356165 while suggesting weaker evidence for rs2583988. Large, multi-ethnic studies are warranted to elucidate underlying mechanisms and support precision medicine in PD.

## 1. Introduction

Parkinson’s Disease (PD) is a prevalent and progressive neurodegenerative disorder affecting millions worldwide, characterized by motor symptoms (e.g., bradykinesia, tremor, rigidity, postural instability) and non-motor symptoms (e.g., cognitive impairment, mood disorders, autonomic dysfunction) [1]. These manifestations lead to substantial morbidity and reduced survival. PD multifactorial etiology involves environmental and genetic factors, with genetic susceptibility increasingly recognized as a modulator of both risk and progression [2,3,4,5].

A key pathological hallmark of PD and Lewy body dementia is the aggregation of the alpha-synuclein protein, which is encoded by the *SNCA* gene, ultimately leading to the formation of Lewy bodies. Rare *SNCA* mutations identified through linkage studies cause familial PD via amino acid substitutions and altered protein configurations [1,6,7]. Beyond these, common *SNCA* variants, including single-nucleotide polymorphisms (SNPs) and promoter region polymorphisms such as the dinucleotide repeat sequence (REP1), have been associated with sporadic PD in various populations [8]. Notably, longer REP1 alleles correlate with increased *SNCA* expression and higher PD risk, reinforcing *SNCA*’s pivotal role in PD pathogenesis [9,10].

Emerging evidence suggests that *SNCA* variants may not only influence PD risk but also act as disease modifiers, affecting onset age, progression, and specific clinical manifestations [11]. For example, in a Han Chinese cohort, the *SNCA* SNPs rs11931074, rs7684318, and rs356219 showed strong linkage disequilibrium. The GG genotype of rs11931074 was associated with reduced PD risk but a significantly increased risk of developing dementia in PD [12]. Similarly, *SNCA* rs6826785 was associated with cognitive decline—non-carriers had a higher risk for PD-mild cognitive impairment (PD-MCI)—highlighting its potential as a therapeutic target for cognitive symptoms [13]. Beyond single variant effects, gene–gene and gene–environment interactions likely modulate PD risk and phenotype [9,14,15,16]. However, while genome-wide association studies (GWAS) have identified over 40 PD-associated loci, including *SNCA*, these explain only a modest portion of heritability. A genome-wide complex trait analysis (GCTA) suggests common variants account for up to 27% of PD risk, leaving much of the genetic contribution unidentified [17].

Given the substantial evidence linking *SNCA* polymorphisms with PD risk and progression, a systematic review and meta-analysis are essential to synthesize existing findings. This study aims to assess the impact of *SNCA* variants on PD susceptibility by integrating primary research studies. By systematically reviewing the literature and conducting meta-analyses on key *SNCA* polymorphisms, we aim to clarify their role in PD pathophysiology, identify inconsistencies, and inform future research. Results may enhance genetic risk prediction and inform therapeutic strategies in PD.

## 2. Methods 

### 2.1. Study Selection

A comprehensive literature search using the electronic databases of PubMed, Scopus, and Web of Science for studies published before 20 November 2023, was conducted. Search terms included the following: (“Parkinson Disease” OR “Parkinson’s Disease”) AND (“Association” OR “Relationship” OR “Correlation” OR “Risk”) AND (“α-Synuclein” OR “*SNCA*” OR “Alpha-Synuclein” OR “aSyn”) AND (“Polymorphisms” OR “Variants” OR “Mutations” OR “Variation”). Four investigators (R.M, M.S, SF.M, and M.Z.Y.C.L)) independently screened studies applying the following inclusion criteria: (1) cohort or case–control studies on *SNCA* variants and PD, (2) English language, (3) human subjects, and (4) sufficient frequency data. Disagreements were resolved by consensus.

### 2.2. Data Extraction

Two researchers (SF.M and C.L.S) independently extracted the following from each study: first author, year of publication, region, ethnicity, sample size, age of case and control, PD type, SNP type, comparison type, odds ratios (ORs), and confidence intervals (CIs). Disagreements were resolved by consensus. Authors were not contacted for additional information.

### 2.3. Quality Assessment

Two investigators (R.M and S.E.S) independently assessed study quality using the Newcastle–Ottawa Scale (NOS) [18], which evaluates selection (4 points), comparability (2 points), and exposure/outcome (3 points), with a maximum of 9. Studies scoring ≥ 7 were rated high quality. Discrepancies were resolved by consensus.

### 2.4. Statistical Analysis

Meta-analyses were performed to evaluate the association between each SNP variant and PD risk, with ORs and 95% CIs calculated as effect sizes. Unadjusted ORs were used for most studies, except for two in which genotype frequencies were unavailable; in these cases, adjusted ORs were utilized. Heterogeneity across studies was assessed using Cochran’s Q test and the I^2^ statistic, where I^2^ values of 25%, 50%, and 75% were considered to indicate low, moderate, and high heterogeneity, respectively. Fixed-effects models were applied when heterogeneity was low (I^2^ < 25%), while random-effects models were used for moderate to high heterogeneity. To explore potential sources of heterogeneity, subgroup meta-analyses based on geographic region were conducted. Sensitivity analysis was performed using the leave-one-out method to examine the influence of individual studies on the pooled effect size. Publication bias was assessed using the trim-and-fill method to identify potential missing studies and evaluate funnel plot asymmetry. All statistical analyses were conducted using R software 4.4.2. (R Core Team (2024); R: A Language and Environment for Statistical Computing. R Foundation for Statistical Computing, Vienna, Austria. https://www.R-project.org). A *p*-value < 0.05 was considered statistically significant.

## 3. Results

The systematic search identified 4665 studies, which were screened after removing duplicates, leaving 3334 studies for abstract screening. Based on the inclusion criteria, 381 studies were selected for full-text screening. However, 22 studies were unavailable, reducing the number to 359. Studies that lacked control groups, focused on Parkinson’s disease dementia (PDD) or early onset PD, or did not provide relevant data were excluded. Ultimately, 44 studies met the final inclusion criteria. Among them, 17 studies did not provide sufficient data on frequencies, while 27 contained sufficient data for meta-analysis. The included studies reported data on four *SNCA* polymorphisms: rs356165 (7 studies [2,19,20,21,22,23,24]; 8 allelic models: G vs. A, 5 dominant: AG + GG vs. AA, and 5 recessive: GG vs. AG + AA), rs2583988 (6 studies [9,19,20,25,26,27]; 7 allelic models: T vs. C, 3 dominant: TC + TT vs. CC, and 3 recessive: TT vs. TC + CC), rs356219 (16 studies [9,16,20,21,24,25,26,28,29,30,31,32,33,34,35,36]; 16 allelic models: G vs. A, 14 dominant: AG + GG vs. AA, and 14 recessive: GG vs. AG + AA), and rs11931074 (13 studies [2,5,19,21,23,24,25,26,37,38,39,40,41]; 14 allelic models: T vs. G, 9 dominant: TT + TG vs. GG, and 8 recessive: TT vs. TG + GG). All included studies were case–control in design, except for one study [28], which was longitudinal but used *SNCA* genotyping at baseline, making it comparable for meta-analysis. The study selection process is illustrated in Figure 1. The detailed characteristics and methodological quality of the included studies are summarized in the Appendix A. Seventeen studies scored 7 or above according to the NOS, indicating high quality and low risk of bias. Ten studies received scores of 5 or 6, suggesting moderate quality (Appendix A). The meta-analysis results for allele and genotype associations of the *SNCA* polymorphisms are presented in Table 1.

### 3.1. SNP rs356165

The meta-analysis of SNP rs356165 under the allelic model (G vs. A) included eight studies [2,19,20,21,22,23,24], revealing a significant association between the G allele and increased PD risk (OR = 1.25, 95% CI: 1.06–1.46, *p*-value = 0.0071; I^2^ = 80.1%; Figure 2A). Leave-one-out sensitivity analyses confirmed the robustness of the association (OR range: 1.22–1.31). Notably, exclusion of study [21], which combined two datasets (PPMI-European and Washington University), eliminated heterogeneity (I^2^ = 0%) and increased precision (OR = 1.31, 95% CI: 1.22–1.41), indicating dataset-specific influence. The trim-and-fill method imputed two missing studies; adjusted results remained significant (OR = 1.20, 95% CI: 1.04–1.38, *p*-value = 0.0136). Subgroup meta-analysis by geographic region showed significant associations in Europe (OR = 1.40, 95% CI: 1.26–1.55) and East Asia (OR = 1.24, 95% CI: 1.12–1.37), while the “Other” category (study [21]) showed an inverse association (OR = 0.77, 95% CI: 0.65–0.92). The subgroup difference was significant (Q = 34.12, df = 2, *p*-value < 0.0001; Figure 2B), suggesting regional variation contributed to heterogeneity.

Under the dominant model (AG + GG vs. AA), five studies [2,20,22,23,24] showed a significant association (OR = 1.37, 95% CI: 1.21–1.54, *p*-value < 0.0001; I^2^ = 7.8%; Appendix A). The sensitivity analysis confirmed stability (OR range: 1.31–1.43), with no publication bias detected. Due to low heterogeneity and limited studies, no subgroup analysis was performed. Similarly, the recessive model (GG vs. AG + AA), using the same five studies [2,20,22,23,24], showed a significant association (OR = 1.50, 95% CI: 1.31–1.71, *p*-value < 0.0001; I^2^ = 0%; Appendix A). The sensitivity analysis and publication bias assessments supported consistent findings.

### 3.2. SNP rs2583988

The meta-analysis of rs2583988 under the allelic model (T vs. C) included seven studies [9,19,20,25,26,27] and showed a borderline significant association with PD risk (OR = 1.22, 95% CI: 1.00–1.48, *p*-value = 0.0446; I^2^ = 79.2%; Figure 3A). However, the sensitivity analysis revealed that this association was not robust, with most leave-one-out iterations yielding non-significant results. Only when Myhre et al. [20] or Trotta et al. [9] were excluded did the association remain significant, suggesting stabilizing effects from these studies. After adjusting for potential publication bias using the trim-and-fill method, the association became non-significant (OR = 1.07, 95% CI: 0.88–1.30, *p*-value = 0.524), indicating possible small-study effects. Subgroup analysis by region showed non-significant results in Europe (OR = 1.13, 95% CI: 0.90–1.41; I^2^ = 75.6%), while significant associations were found in single studies from the U.S. (OR = 1.42, 95% CI: 1.18–1.71) and Brazil (OR = 1.68, 95% CI: 1.06–2.67). However, the test for subgroup differences was not statistically significant (Q = 3.57, df = 2, *p*-value = 0.1681; Figure 3B), suggesting that the geographic region does not fully account for heterogeneity.

Under the dominant model (TT + TC vs. CC), based on three studies [20,25,26], no significant association was observed (OR = 1.14, 95% CI: 0.93–1.40, *p*-value = 0.208; I^2^ = 0%; Appendix A), and findings remained non-significant across the sensitivity and publication bias analyses. Similarly, the recessive model (TT vs. TC + CC) showed no significant association (OR = 1.31, 95% CI: 0.86–2.01, *p*-value = 0.206; I^2^ = 55.2%; Appendix A), with instability noted in the sensitivity analysis and no evidence of a publication bias.

### 3.3. SNP rs356219

The meta-analysis of rs356219 under the allelic model (G vs. A), including 16 studies [9,16,20,21,24,25,26,28,29,30,31,32,33,34,35,36], showed a significant association with increased PD risk (OR = 1.35, 95% CI: 1.22–1.50, *p*-value < 0.0001; I^2^ = 80.7%; Figure 4A). The leave-one-out sensitivity analyses confirmed robustness (OR range: 1.32–1.40), and the association remained significant after adjusting for publication bias (OR = 1.23, 95% CI: 1.10–1.38, *p*-value = 0.0003), despite imputation of five potentially missing studies. A subgroup analysis by region revealed significant associations across all areas, with the strongest effects in South and Central America (ORs = 1.70 and 1.71, respectively), though based on single studies. Among regions with multiple studies, East Asia showed the most consistent association (OR = 1.50, 95% CI: 1.28–1.76), followed by Combined (OR = 1.36, 95% CI: 1.23–1.50) and Europe (OR = 1.18, 95% CI: 1.01–1.38). However, the test for subgroup differences was not significant (Q = 7.12, df = 4, *p*-value = 0.1296; Figure 4B).

Under the dominant model (AG + GG vs. AA), 14 studies [16,20,24,25,26,28,29,30,31,32,33,34,35,36] showed a significant association (OR = 1.46, 95% CI: 1.29–1.65, *p*-value < 0.0001; I^2^ = 51.8%; Appendix A). The sensitivity analysis confirmed consistency, and the association remained significant after adjusting for three potentially missing studies (OR = 1.36, 95% CI: 1.20–1.55).

The recessive model (GG vs. AG + AA), using the same 14 studies, revealed a stronger association (OR = 1.69, 95% CI: 1.49–1.92, *p*-value < 0.0001; I^2^ = 50.2%; Appendix A). The result was stable in the sensitivity analysis and remained significant after adjusting for five potentially missing studies (OR = 1.49, 95% CI: 1.29–1.72). The subgroup analyses for both models showed consistent associations across all regions. In the dominant model, East Asia showed the strongest effect (OR = 1.65, 95% CI: 1.34–2.03), followed by Combined (OR = 1.55, 95% CI: 1.28–1.88) and Europe (OR = 1.26, 95% CI: 1.10–1.43); subgroup differences were not significant (Q = 7.17, df = 4, *p*-value = 0.1273; Appendix A). Similar trends were observed under the recessive model, with East Asia again showing the highest effect (OR = 1.75, 95% CI: 1.39–2.22), followed by Europe (OR = 1.54, 95% CI: 1.34–1.78) and Combined (OR = 1.53, 95% CI: 1.26–1.86); subgroup differences were also non-significant (Q = 3.44, df = 4, *p*-value = 0.4863; Appendix A).

### 3.4. SNP rs11931074

A meta-analysis of 14 studies [2,5,19,21,23,24,25,26,37,38,39,40,41] under the allelic model (T vs. G) showed a significant association between the T allele and increased PD risk (OR = 1.36, 95% CI: 1.30–1.42, *p*-value < 0.0001; I^2^ = 13.7%; Figure 5A), with low heterogeneity and no evidence of publication bias. The leave-one-out sensitivity analyses confirmed the stability of the results (OR range: 1.33–1.37). Given the consistency, no subgroup analysis was conducted.

Under the dominant model (TT + TG vs. GG), based on nine studies [2,23,24,25,26,37,38,40,41], a significant association was observed (OR = 1.49, 95% CI: 1.35–1.66, *p*-value < 0.0001; I^2^ = 0%), with robust sensitivity results (OR range: 1.46–1.53) and no publication bias (Figure 5B). The recessive model (TT vs. TG + GG), including eight studies [2,23,24,25,37,38,40,41], also showed a significant association (OR = 1.48, 95% CI: 1.28–1.70, *p*-value < 0.0001; I^2^ = 36.3%; Figure 6A). The results remained consistent in the sensitivity analysis (OR range: 1.43–1.54), and no missing studies were identified via trim-and-fill. The subgroup analysis for the recessive model showed the strongest association in East Asia (4 studies, OR = 1.53, 95% CI: 1.36–1.72). Associations were weaker and non-significant in the Middle East (2 studies), South America (1 study), and Europe (1 study), though interpretation was limited by small sample sizes. No significant subgroup differences were detected (Q = 1.09, df = 3, *p*-value = 0.7803; Figure 6B).

## 4. Discussion

PD is a complex neurodegenerative disorder, with growing evidence implicating genetic variability in its susceptibility and progression [42]. A hallmark feature of PD is the presence of Lewy bodies, which are intracellular inclusions primarily composed of aggregated alpha-synuclein protein. The alpha-synuclein protein is encoded by the *SNCA* gene located on chromosome 4q22 [43,44]. While *SNCA* mutations are rare and typically associated with familial PD, common variants have been shown to influence the risk of sporadic PD. Additionally, extracellular alpha-synuclein levels have been linked to PD diagnosis and severity, supporting its role as a potential biomarker.

Over the past two decades, GWASs have identified over 40 loci significantly associated with PD, highlighting the role of genetic factors in disease pathogenesis [45]. Among these, *SNCA* has emerged as a key locus due to its central involvement in Lewy body formation and regulation of alpha-synuclein expression. Evidence suggests that *SNCA* variants can influence both protein levels and transcript diversity, thereby modulating PD risk [13,38,46,47,48,49]. In this context, the present study aimed to systematically evaluate and synthesize existing evidence on the association between *SNCA* polymorphisms and PD susceptibility through a comprehensive meta-analysis.

In this comprehensive meta-analysis, we evaluated the association between four *SNCA* single-nucleotide polymorphisms—rs356165, rs2583988, rs356219, and rs11931074—and PD risk. Among the 27 studies included, our results confirmed that rs356165, rs356219, and rs11931074 are significantly associated with increased PD risk across allelic, dominant, and recessive genetic models. Notably, rs356219 demonstrated the strongest association, particularly under the recessive model (OR = 1.69, 95% CI: 1.49–1.92). In contrast, rs2583988 showed a weaker and less consistent association, reaching marginal significance only under the allelic model and not under dominant or recessive models. Subgroup analyses by geographic region revealed generally consistent effects, with particularly strong associations observed in East Asian populations. Our findings align with those of previous large-scale meta-analyses investigating *SNCA* polymorphisms. Specifically, a comprehensive meta-analysis by Zhang et al. (2018) identified eight *SNCA* SNPs associated with PD risk, including rs356165, rs356219, and rs11931074—three of the four SNPs evaluated in our study [50]. These variants were also among those with the strongest associations in Zhang et al.’s analysis, with rs11931074 and rs356219 demonstrating increased risk across multiple ethnic groups, including East Asian and European populations. The consistency between our findings and those of Zhang et al. strengthens the evidence supporting these *SNCA* variants as robust genetic risk factors for PD and highlights their potential relevance in genetic screening strategies.

In the following sections, we discuss the specific findings and implications of each *SNCA* variant in detail.

### 4.1. SNP rs356165

Our meta-analysis consistently demonstrates that SNP rs356165 is associated with increased PD risk across allelic, dominant, and recessive models. Heterogeneity in the allelic model was primarily driven by one outlier study; its exclusion improved consistency. Subgroup analyses confirmed significant associations in both European and East Asian populations, though the combined dataset showed an inverse effect, likely due to population-specific differences. The dominant and recessive models yielded stable, significant associations with minimal heterogeneity and no publication bias.

Located in the 3′ untranslated region (3′ UTR) of *SNCA*, rs356165 plays a regulatory role in post-transcriptional expression. Ghanbari et al. [51] proposed that the G allele disrupts microRNA binding sites, potentially increasing *SNCA* expression. This aligns with findings of elevated *SNCA* mRNA and α-synuclein protein levels in G allele carriers versus AA genotypes [42]. These effects may be especially pronounced in brain tissue, where longer 3′ UTRs amplify post-transcriptional regulation. Additional studies further support rs356165’s biological relevance. A Spanish study reported significant association under the dominant model and with earlier age at onset (AAO) [52], while a Chinese study found no significant association [53], indicating possible ethnic differences. Rajput et al. [54] validated the association in a Canadian cohort.

Regarding related phenotypes, a Croatian study found no direct association between rs356165 and PD but reported intermediate linkage disequilibrium with other *SNCA* 3′ UTR variants (rs1045722 and rs3857053), suggesting shared regulatory mechanisms [42]. A population-based study suggested an additive interaction between head injury and rs356165, with a 3–4.5-fold increased PD risk in carriers post-trauma, though no multiplicative interaction was found [55]. Moreover, rs356165 was more frequent in PD than in idiopathic REM sleep behaviour disorder (iRBD), implying a potential role in prodromal disease progression [56]. A longitudinal study also showed that while rs356165 alone did not predict motor progression, it contributed additively when combined with the REP1 promoter variant [57].

In sum, our meta-analysis and supporting literature provide strong evidence for rs356165’s role in PD susceptibility. Its consistent associations across genetic models, functional implications in *SNCA* regulation, and links to phenotypic variability and progression underscore its biological significance. Future studies involving functional assays and diverse populations are warranted to further clarify its role in PD pathogenesis.

### 4.2. SNP rs2583988

Our meta-analysis indicates a possible but inconclusive association between rs2583988 and PD risk. A marginally significant effect was observed under the allelic model; however, this association lacked robustness due to substantial heterogeneity and inconsistent sensitivity analysis results. After adjusting for publication bias, the effect was no longer significant, suggesting small-study influences. Subgroup analysis by region showed no association in European cohorts, while stronger effects were reported in single studies from the United States and Brazil, limiting generalizability. Neither the dominant nor recessive models showed significant associations, indicating that any link between rs2583988 and PD is likely weak and uncertain.

Beyond our analysis, one study found that individuals homozygous for the C allele had a higher risk of rapid motor progression, though this association did not remain significant after correction for multiple comparisons [54]. The overall heterogeneity and methodological differences across studies underscore the need for larger, well-powered investigations to clarify the potential role of rs2583988 in PD susceptibility and progression.

### 4.3. SNP rs356219

Our meta-analysis provided strong and consistent evidence for an association between SNP rs356219 and increased PD risk across all genetic models. The G allele was significantly linked to higher PD risk, with results remaining stable in sensitivity and publication bias analyses. While heterogeneity was notable under the allelic model, subgroup meta-analyses confirmed the association across all regions, particularly in East Asian populations. The dominant and recessive models also supported this association, with consistent findings across studies and minimal heterogeneity. These findings position the rs356219 G allele, especially in homozygous form, as a robust genetic risk factor for PD across diverse populations. These findings are in line with the previous systematic review and meta-analysis by Hou et al. (2019), which also reported significant associations between rs356219 and PD risk across all genetic models [58].

Subgroup comparisons showed regional variability in effect size, though not statistically significant. The strongest associations were observed in East Asia, followed by combined and European datasets. These findings align with studies from Chinese populations that highlight *SNCA* as a key contributor to PD susceptibility [9,14,16,30,33,59]. For instance, a large Han Chinese case–control study reported significant associations between the G allele and increased PD risk and earlier AAO [33]. In contrast, Refenes et al. found no association in Greek and Italian cohorts, suggesting ethnic variability [60]. In Japan, rs356219 was not directly associated with PD risk, but a significant additive interaction with smoking was observed [34].

Functional studies support the biological relevance of rs356219. The G allele has been associated with elevated α-synuclein mRNA and protein levels, and its combination with plasma α-synuclein improved early PD diagnosis [31]. Szwedo et al. [28] reported faster cognitive decline in GG carriers, and Goris et al. [32] observed a synergistic effect between rs356219 and MAPT variants in increasing dementia risk. Emelyanov et al. [26] further demonstrated increased α-synuclein in CD45+ blood cells of G allele carriers, indicating regulatory effects on *SNCA* expression.

Additional studies highlight rs356219 as a disease-modifying variant. In a Spanish LRRK2-PD cohort, the GG genotype was associated with PD onset up to 11 years earlier than the AA genotype [61]. This was supported by regression models showing consistent associations between the risk allele and earlier PD onset [62] and by studies reporting gene–gene interactions between rs356219 and the LRRK2 G2019S mutation [63]. A 30-month longitudinal study also found the G allele linked to reduced motor and cognitive decline risk, especially among rapid eye movement behaviour disorder (RBD) patients, suggesting a potential prognostic role [64].

Complementary evidence has emerged from studies examining rs356219 under the C vs. T model [65,66,67]. Mata et al. (2010) found the C allele to be a stronger predictor of PD risk than REP1 in U.S. cohorts and associated it with elevated plasma α-synuclein under an additive model [65]. Lucchini et al. (2020) reported significantly higher PD risk among CC genotype carriers in an Italian cohort, with additive—but not multiplicative—effects in individuals exposed to metals [68].

Collectively, these findings confirm rs356219 as a strong susceptibility and disease-modifying marker in PD. Its impact on *SNCA* expression, cognitive decline, and interactions with environmental and genetic factors underscores its multifaceted role in PD pathogenesis, particularly in East Asian populations. Future research should prioritize integrative approaches to clarify its clinical and biological implications across diverse populations.

### 4.4. SNP rs11931074

Our meta-analysis demonstrated a consistent and significant association between SNP rs11931074 and increased PD risk across allelic, dominant, and recessive models. These associations remained robust in sensitivity analyses, with low heterogeneity and no evidence of publication bias. Subgroup analyses confirmed the association across regions, particularly in East Asian and European populations, supporting rs11931074 as a reliable genetic marker of PD susceptibility.

Located 7.2 kb downstream of the 3′ end of the *SNCA* gene, rs11931074 has been repeatedly implicated in PD risk [69]. Despite notable differences in allele frequency between populations, where the T allele is more common in Asians (~58%) and less frequent in Caucasians (~7%), an overrepresentation of the T allele among PD patients has been consistently observed across diverse ethnic groups [40,70]. Our findings align with prior GWAS and case–control studies, including one in European populations identifying rs11931074 as a top PD risk variant [4], and others in Chinese and Korean cohorts reporting associations with both disease susceptibility and milder clinical phenotypes, such as hyposmia and less severe motor symptoms [71,72].

Beyond risk association, rs11931074 has been implicated in PD heterogeneity. The T allele is linked to lower serum α-synuclein levels and non-motor symptoms like constipation and REM sleep behaviour disorder [40,72]. TT genotype carriers often exhibit milder motor symptoms, slower progression, and lower comorbidity burden [73,74]. Additional studies report reduced salivary α-synuclein and altered brain activity in TT carriers, specifically increased right angular gyrus activity inversely correlated with motor severity [75,76].

Although some studies report conflicting findings, such as a possible protective role of the G allele in early-onset or familial PD [77] or no association with peripheral α-synuclein levels [26], the broader evidence supports a multifactorial role for rs11931074 in PD risk and phenotype.

Functionally, rs11931074 affects *SNCA* transcript processing, increasing the expression of shorter isoforms like *SNCA*-98 and *SNCA*-112, which may enhance protein aggregation [22]. Our findings are further supported by a recent large-scale meta-analysis of 13 studies involving over 13,000 PD cases and 28,000 controls, which confirmed a robust association between rs11931074 and PD risk across five genetic models, with the strongest effects observed in Asian populations [78]. Furthermore, emerging evidence suggests that rs11931074 may also influence peripheral α-synuclein pathology. Moreover, exploratory research in the enteric nervous system (ENS) found the G allele associated with increased α-synuclein immunostaining (OR = 5.96, *p* = 0.01) and a significant interaction with PD status, suggesting potential involvement in early or extracerebral PD pathology [79].

Taken together, these findings reinforce rs11931074 as a key susceptibility variant that may also influence the PD phenotype and progression, potentially through mechanisms involving *SNCA* regulation and α-synuclein expression.

## 5. Strengths and Limitations

This meta-analysis synthesizes data from 44 studies, with 27 providing sufficient quantitative data for analysis, leveraging robust statistical methods like leave-one-out sensitivity analysis and trim-and-fill to ensure finding stability. It systematically evaluates four *SNCA* polymorphisms across genetic models, yielding consistent associations for rs356165, rs356219, and rs11931074.

A key strength of our study is its timeliness and comprehensiveness. Unlike previous systematic reviews, which were published before 2019, our review includes studies up to 20 November 2023, allowing us to incorporate a broader and more current evidence base. We also explicitly acknowledged the existence of previous reviews and outlined clear differences in our search timeline, number of included studies, and variants analyzed. These distinctions enhance the relevance and clinical utility of our findings in the current research landscape. However, limitations include the exclusion of 17 studies due to inadequate genotype or OR data, potentially reducing generalizability. Notable heterogeneity, especially in allelic models of rs356165 (I^2^ = 80.1%) and rs356219 (I^2^ = 80.7%), suggests variability from study design, population differences, or unmeasured factors (e.g., genotyping accuracy, environmental exposures). Additionally, the geographic distribution of studies is skewed toward European and East Asian populations, with limited representation from other ethnic groups, which may restrict the global generalizability of the findings.

## 6. Implications and Future Directions

This meta-analysis highlights the contribution of *SNCA* polymorphisms—particularly rs356165, rs11931074, and rs356219—to PD risk across genetic models and populations. These variants may aid in polygenic risk stratification and selection for α-synuclein-targeted therapies. Their regulatory effects also suggest potential as progression biomarkers. Heterogeneity observed in rs356165, rs2583988, and rs356219 underscores the need for larger, multi-ethnic studies, especially in underrepresented African and South American populations. Integrating genetic, environmental, and clinical data is essential to understanding PD’s etiology. As common variants explain only ~27% of PD heritability [17], polygenic models, functional studies, and longitudinal cohorts are needed to clarify variant effects on disease onset and progression.

## 7. Conclusions

This meta-analysis provides strong evidence that *SNCA* variants—particularly rs11931074—are robustly associated with increased PD risk across diverse populations and genetic models, with consistent findings, low heterogeneity, and no publication bias. Rs356165 and rs356219 also showed significant associations across all models, though high heterogeneity limits interpretability. In contrast, rs2583988 demonstrated weaker, context-dependent associations with greater variability across studies. Despite these limitations, our findings reinforce *SNCA*’s central role in PD pathogenesis and underscore the importance of future large-scale, ethnically diverse studies to explore gene–environment and gene–gene interactions and support the development of personalized therapeutic strategies.

## Figures and Tables

**Figure 1 ijms-26-06001-f001:**
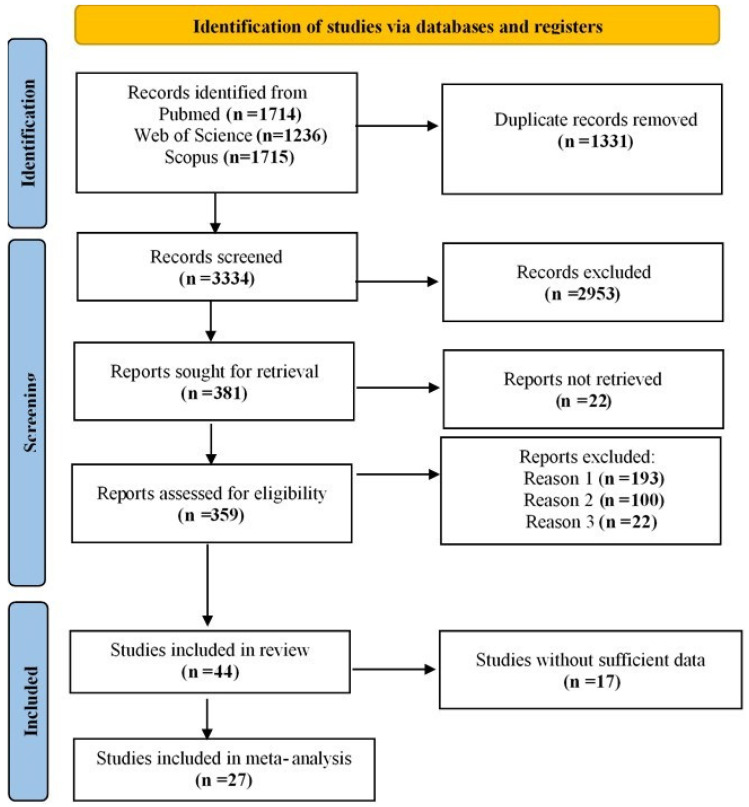
PRISMA flow diagram illustrating the study selection process.

**Figure 2 ijms-26-06001-f002:**
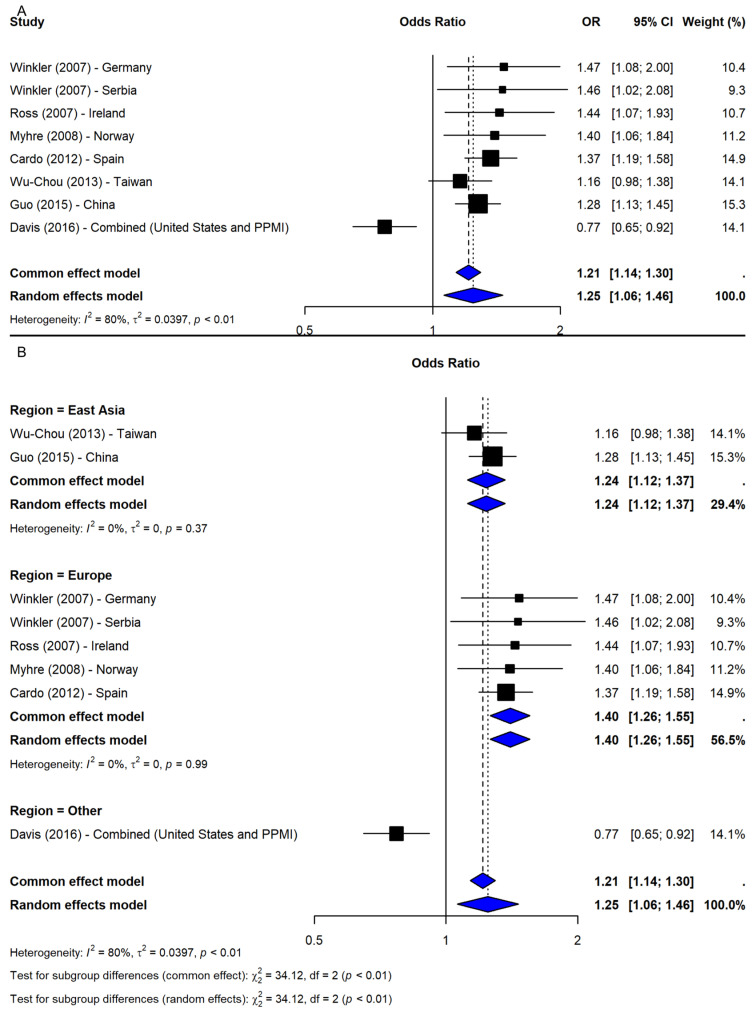
Association between SNP rs356165 and PD risk under allelic model (G vs. A) [2,19,20,21,22,23,24]: (**A**) overall meta-analysis; (**B**) subgroup analyses by geographic region. Black squares represent odds ratios (ORs) from individual studies, with their size reflecting study weight; horizontal lines indicate 95% confidence intervals. Blue diamonds represent the pooled ORs.

**Figure 3 ijms-26-06001-f003:**
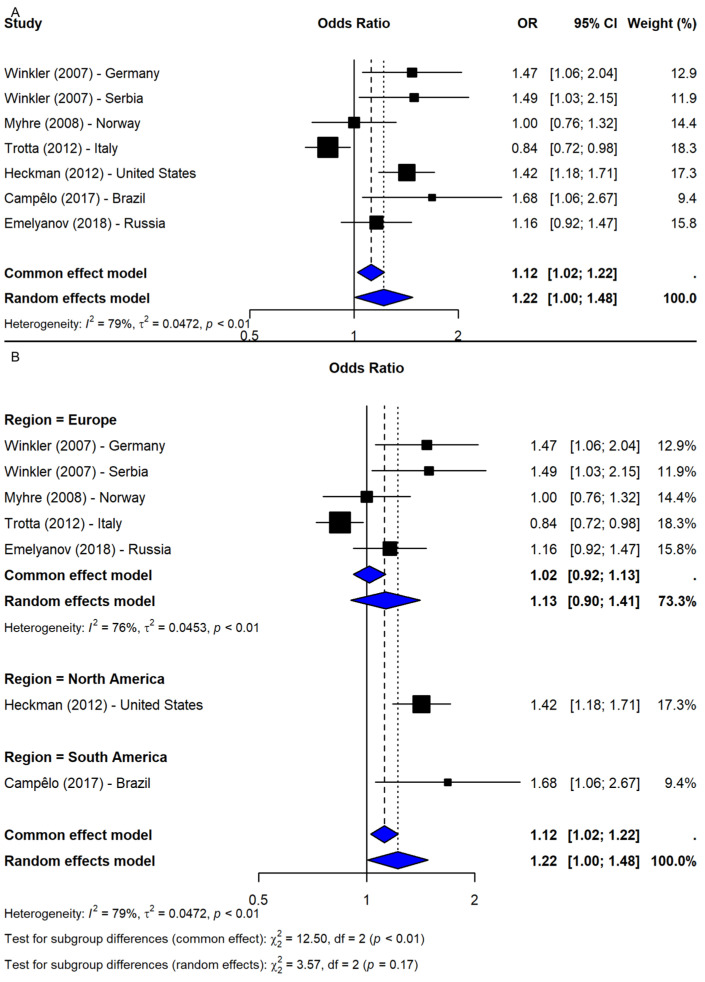
Association between SNP rs2583988 and PD risk under allelic model (T vs. C) [9,19,20,25,26,27]: (**A**) overall meta-analysis; (**B**) subgroup analyses by geographic region. Black squares represent ORs from individual studies, with their size reflecting study weight; horizontal lines indicate 95% confidence intervals. Blue diamonds represent the pooled ORs.

**Figure 4 ijms-26-06001-f004:**
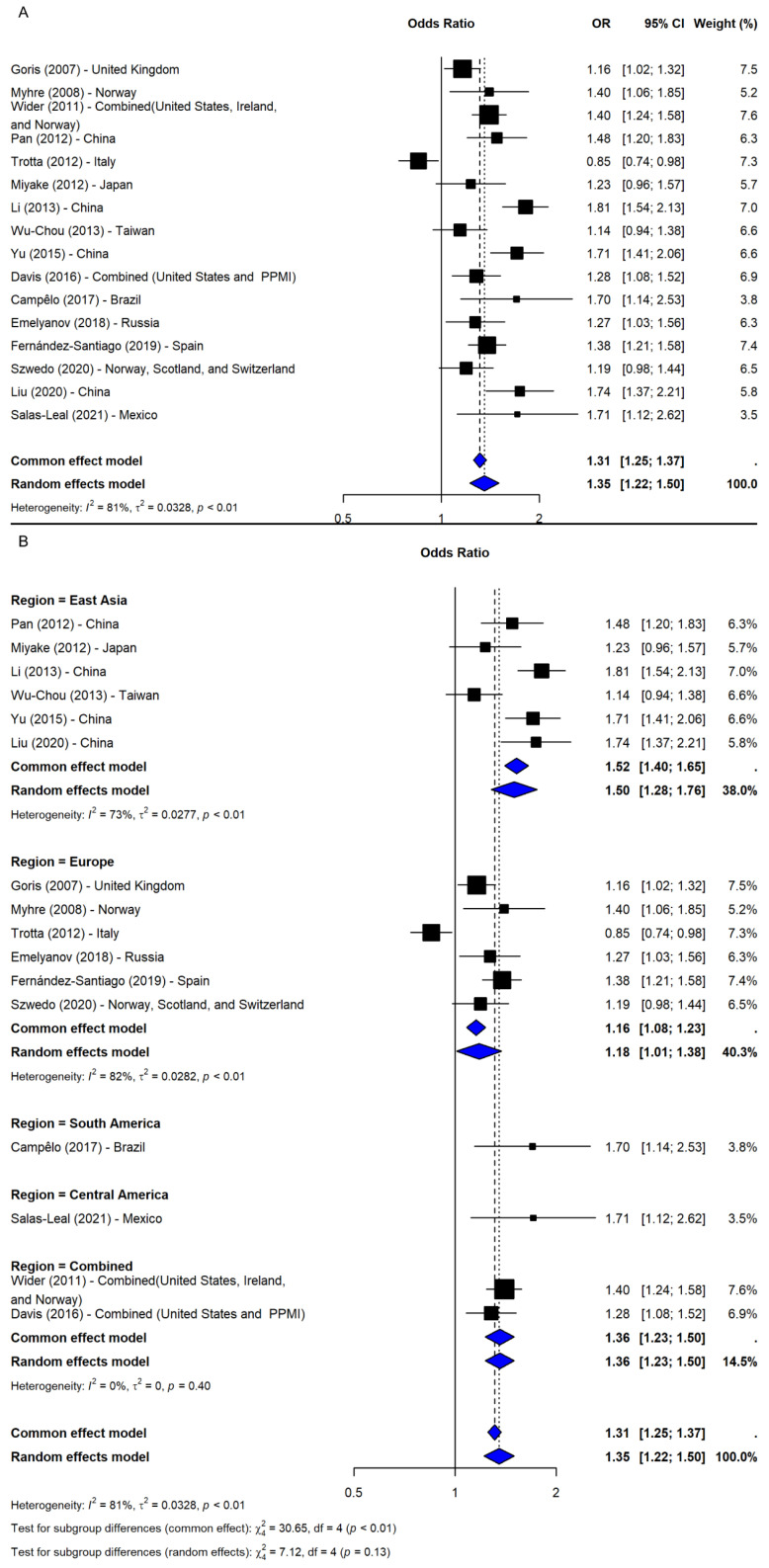
The association between SNP rs356219 and PD risk under the allelic model (G vs. A) [9,16,20,21,24,25,26,28,29,30,31,32,33,34,35,36]: (**A**) the overall meta-analysis; (**B**) the subgroup analyses by geographic region. Black squares represent ORs from individual studies, with their size reflecting study weight; horizontal lines indicate 95% confidence intervals. Blue diamonds represent the pooled ORs.

**Figure 5 ijms-26-06001-f005:**
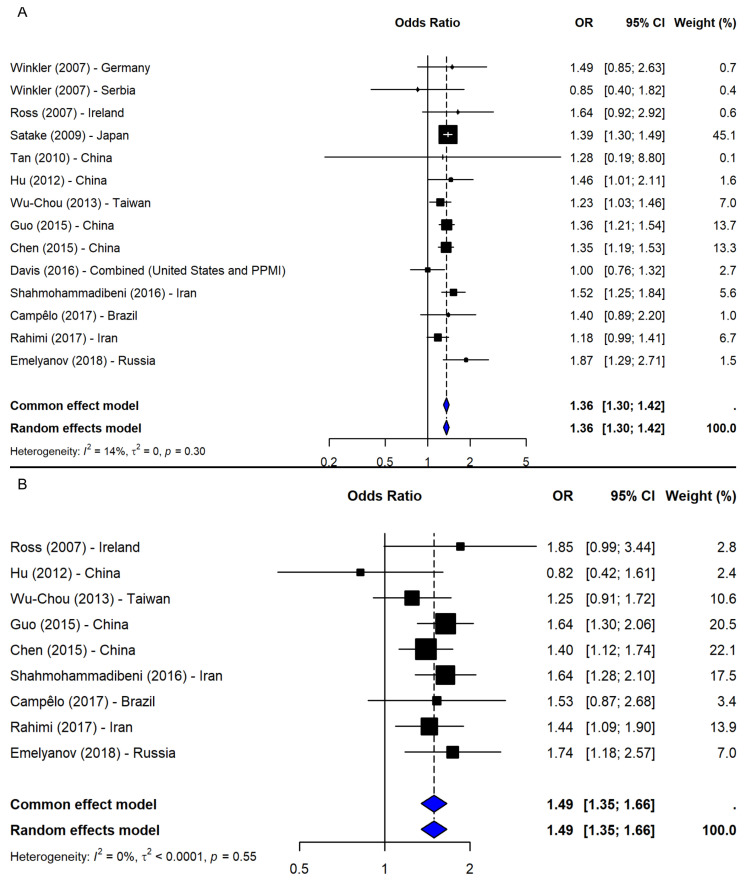
Association between SNP rs11931074 and PD risk under the (**A**) allelic model (T vs. G) [2,5,19,21,23,24,25,26,37,38,39,40,41] and the (**B**) dominant model (TT + TG vs. GG) [2,23,24,25,26,37,38,40,41]. Black squares represent ORs from individual studies, with their size reflecting study weight; horizontal lines indicate 95% confidence intervals. Blue diamonds represent the pooled ORs.

**Figure 6 ijms-26-06001-f006:**
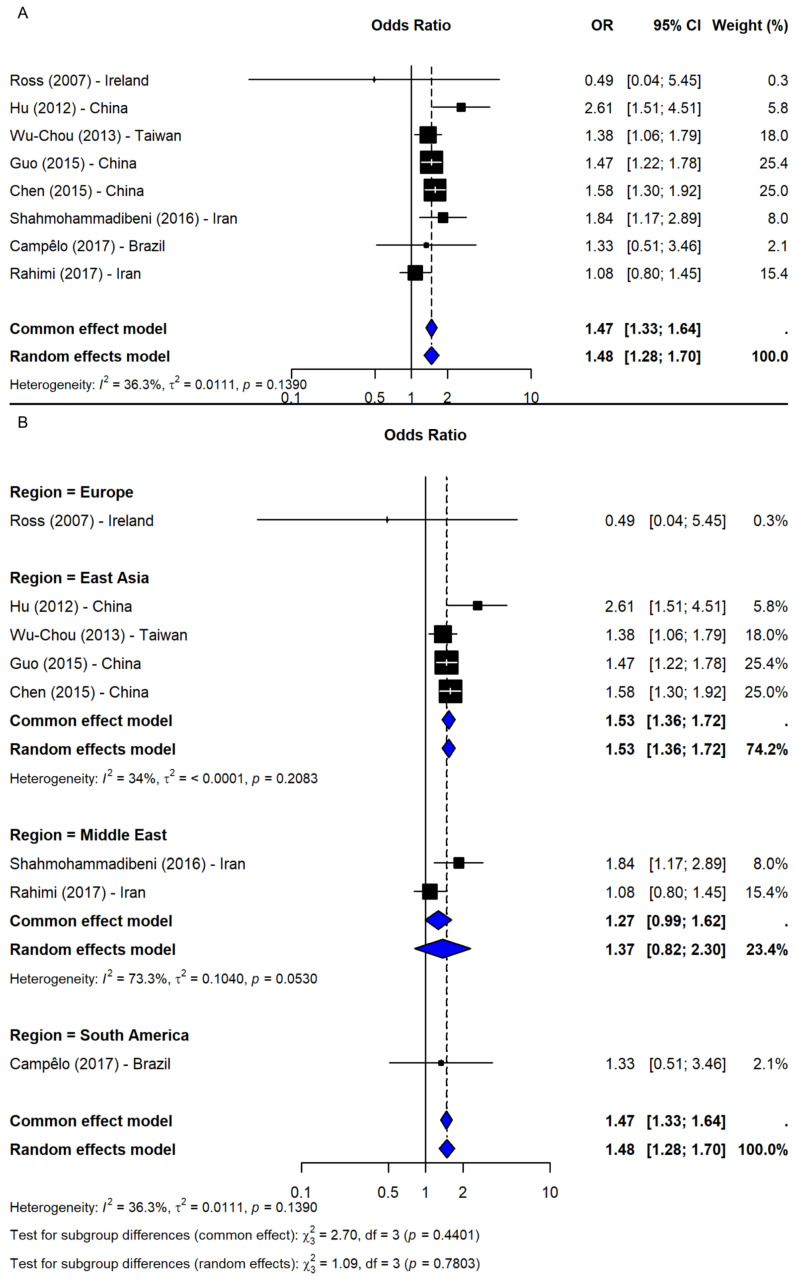
Association between SNP rs11931074 and PD risk under the recessive model (TT vs. TG + GG) [2,23,24,25,37,38,40,41]: (**A**) the overall meta-analysis; (**B**) the subgroup analyses by geographic region. Black squares represent ORs from individual studies, with their size reflecting study weight; horizontal lines indicate 95% confidence intervals. Blue diamonds represent the pooled ORs.

**Table 1 ijms-26-06001-t001:** Summary of meta-analysis results for the association between SNPs and PD risk.

SNP ID	Comparison	N	OR (95% CI)	*p*-Value ^1^	Z	I-Square	*p*-Value ^2^	Model
	Allelic G vs. A	8	1.25 (1.06, 1.46)	0.0071	2.69	80.1%	<0.0001	R
**rs356165**	Dominant AG + GG vs. AA	5	1.37 (1.21, 1.54)	<0.0001	5.17	7.8%	0.3620	F
	Recessive GG vs. AG + AA	5	1.50 (1.31, 1.71)	<0.0001	5.85	0.0%	0.8834	F
	Allelic: T vs. C	7	1.22 (1.00, 1.48)	0.0446	2.01	79.2%	<0.0001	R
**rs2583988**	Dominant TT + TC vs. CC	3	1.14 (0.93, 1.40)	0.2076	1.26	0.0%	0.4163	F
	Recessive TT vs. TC + CC	3	1.31 (0.86, 2.01)	0.2057	1.27	55.2%	0.1070	R
	Allelic: G vs. A	16	1.35 (1.22; 1.50)	<0.0001	5.74	80.7%	<0.0001	R
**rs356219**	Dominant: AG + GG vs. AA	14	1.46 (1.29, 1.65)	<0.0001	6.10	51.8%	0.0125	R
	Recessive: GG vs. AG + AA	14	1.69 (1.49, 1.91)	<0.0001	8.12	50.2%	0.0164	R
	Allelic: T vs. G	14	1.36 (1.30, 1.42)	<0.0001	13.03	13.7%	0.3029	F
**rs11931074**	Dominant: TG + TT vs. GG	9	1.49 (1.35, 1.66)	<0.0001	7.61	0.0%	0.5477	F
	Recessive: TT vs. TG + GG	8	1.48 (1.28,1.70)	<0.0001	5.43	36.3%	0.1390	R

Abbreviations: N: number of included studies; OR: odds ratio; CI: confidence interval; Z: Z-score for statistical significance; I-square: I^2^ statistic for heterogeneity test, Model: meta-analysis model used. R: random-effects model, F: fixed-effects model. ^1^: *p*-value for overall effect. ^2^: *p*-value for heterogeneity among studies.

## Data Availability

The data supporting the findings of this study (i.e., extracted study-level information) are available from the corresponding author upon reasonable request.

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
