# Peer review of "Common SNCA Genetic Variants and Parkinson’s Disease Risk: A Systematic Review and Meta-Analysis"

_ijms, 2025, doi:10.3390/ijms26136001_

Round 1
Reviewer 1 Report
Comments and Suggestions for Authors
The study provides a robust systematic review and meta-analysis of SNCA polymorphisms in Parkinson’s disease (PD), adhering to established guidelines for genetic epidemiology. As we all known, the relation of SNCA polymorphisms and PD has reported. So there is lack of originality.
Reviewer 2 Report
Comments and Suggestions for Authors
This is an interesting systematic review and meta-analysis that investigates the role of SNCA gene, which encodes alpha-synuclein, common variants in Parkinson’s Disease.
Comments
In general, this is a well wtitten paper, following properly the review and meta-analysis methodology and using adequate statistical methods to process the data. The authors also detailed the findings of their study in discussion section and limitations were clearly stated. Finally, figures and plots were well designed and comprehensive. I have only two minor comments to make:
Comment 1. Page 1, line 34: In keywords the authors should add two more words: Parkinson's disease instead of neurodegenerative disorders, as the review concerns only PD patients. Also, I suggest to add the SNCA gene.
Comment 2. Page 2, line 45: The authors should clarify that the hallmark of PD and also of LBD, is the aggregation of alpha-synuclein protein and ultimately the formation of Lewy bodies and not the alpha-synuclein (SNCA) gene (which is not the primary component of Lewy bodies).
Reviewer 3 Report
Comments and Suggestions for Authors
The authors performed a meta-analysis of association between SNCA variants and Parkinson’s Disease. The introduction, methods and results are properly written. I have however two comments.
1) Such data can be very useful for other researchers but there are already several meta-analyses (listed below). The authors should explain what was the reason for performing the new study, what is a difference between the submitted meta-analysis and previously published studies and discuss obtained data in context of previous meta-analyses.
Schierding et al., Machine Learning Identifies Six Genetic Variants and Alterations in the Heart Atrial Appendage as Key Contributors to PD Risk Predictivity. Front Genet. 2022 Jan 3;12:785436. doi: 10.3389/fgene.2021.785436.
Naushad et al., Alpha synuclein (SNCA) rs7684318 variant contributes to Parkinson's disease risk by altering transcription factor binding related with Notch and Wnt signaling. Neurosci Lett. 2021 Apr 17;750:135802. doi: 10.1016/j.neulet.2021.135802.
Du et al., Association between alpha-synuclein (SNCA) rs11931074 variability and susceptibility to Parkinson's disease: an updated meta-analysis of 41,811 patients.Neurol Sci. 2020 Feb;41(2):271-280. doi: 10.1007/s10072-019-04107-8.
Hou et al., Association of rs356219 and rs3822086 polymorphisms with the risk of Parkinson's disease: A meta-analysis. Neurosci Lett. 2019 Sep 14;709:134380. doi: 10.1016/j.neulet.2019.134380.
Zhang et al., A Comprehensive Analysis of the Association Between SNCA Polymorphisms and the Risk of Parkinson's Disease. Front Mol Neurosci. 2018 Oct 25;11:391. doi: 10.3389/fnmol.2018.00391.
2) There is a following information: “The following supporting information can be downloaded at: https://www.mdpi.com/article/doi/s1, Figure S1: title; Table S1: title; Video S1: title.” I haven’t noticed any supplementary video associated with this manuscript.
Round 2
Reviewer 2 Report
Comments and Suggestions for Authors
This revised manuscript has been improved, since raised comments and suggestions have been adequately addressed.